# Knowledge, Attitudes, and Practices of Community Health Workers in Relation to Environmental Health Hazards in the Nelson Mandela Bay

**DOI:** 10.3390/ijerph21030353

**Published:** 2024-03-15

**Authors:** David G. Morton, Mpinane F. Senekane

**Affiliations:** Department of Environmental Health, Faculty of Health Sciences, University of Johannesburg, Johannesburg 2028, South Africa; david.morton@mandela.ac.za

**Keywords:** knowledge, attitudes, practices, community health workers, environmental health hazards, Nelson Mandela Bay

## Abstract

Background: Community health workers (CHWs) cover extensive areas observing the environmental conditions in which community members live. However, current CHW training modules do not have modules focusing specifically on environmental health. CHWs appear to lack knowledge of environmental health hazards, and little is known of their attitudes and practices regarding environmental health hazards. The purpose of this study was to determine the knowledge, attitudes, and practices of CHWs in relation to environmental health hazards in the Nelson Mandela Bay (NMB). Methods: This study used a quantitative, cross-sectional research design. A sample of 110 respondents completed the questionnaire. The questionnaire was based on the literature, consisting of 36 items in four sections. Data analysis consisted of descriptive and inferential statistics. Reliability and validity were enhanced by utilizing a pre-test study. Results: There were significant differences in attitudes (t = −2.308, df = 91.107, *p* = 0.023) and practices (t = −2.936, df = 62.491, *p* = 0.005). Those trained in environmental health had a significantly lower mean attitudes score (m = 3.2365, sd = 1.113) compared to those not trained in environmental health (m = 3.694, sd = 0.894). In addition, those trained in environmental health had a significantly lower mean practice score (practiced more frequently) (m = 1.231, sd = 0.327) compared to those not trained in environmental health (m = 1.4605, sd = 0.4162). Regarding training, 62% (*n* = 67) of CHWs felt they needed additional training in environmental health. Conclusion: Most of the CHWs had a moderate knowledge of environmental health hazards. Furthermore, most of the CHWs had a very positive or positive attitude towards environmental health hazards. However, there is a need for CHWs to receive very specific training in environmental health. In addition, the scope of work of CHWs, as well as their role in relation to environmental health, needs to be further explored.

## 1. Introduction

Environmental health hazards are a major cause of ill health and death in populations around the world. According to the WHO, in 2016, modifiable environmental health factors were the cause of 13.7 million people dying annually (24% of deaths worldwide) and a result of living or working in unsafe or unhealthy environments [1,2]. In a report on pollution and health [3], it was found that pollution was responsible for an estimated nine million deaths (16% of all deaths globally) and for economic losses amounting to US$ 4.6 trillion in 2015. Such statistics demonstrate the impact of environmental health hazards on human health on a global scale.

The WHO Health and Environment 2022 Scorecard demonstrates that South Africa has several challenges regarding environmental health [4]. Regarding air pollution, South Africa has six times the WHO quality guideline value for fine particulate matter. The NDoH (2019) [5] describes indoor air pollution as caused by a range of factors, such as building materials and furnishings (e.g., asbestos), carpets that are wet or damp, wooden furniture that contains formaldehyde, pesticides, certain paint products and cleaning agents, animals, molds, dust mites, and even tobacco smoke. In addition, 14% of the population does not have access to clean fuels and technology for cooking. Regarding water and sanitation, 51% of deaths from diarrhea are a result of unsafe drinking water, sanitation, and inadequate personal hygiene. Furthermore, in relation to chemical hazards, a concerning 6 out of 100,000 children under five die from poisonings every year in South Africa, with South Africa ranked 23rd out of 47 countries in Africa [6]. In addition, it is estimated that in South Africa, 124 out of every 100,000 deaths of children under five are linked to the environment [7].

An Eastern Cape cross-sectional study on air pollution in schools [8] found that levels of particulate matter in classrooms were above the WHO daily limits and were negatively associated with lung function in the children under study. Another study looked at environmental exposure factors on child diarrhea in the Eastern Cape and found that the transmission of diarrhea was enhanced where multiple households utilized the same sanitation facilities [9]. Hence, the study illustrated that housing conditions exacerbated the health of the communities. Similarly, another study highlighted that in the Eastern Cape, there is a major shortage of quality housing, which has health implications for many of the people living in the province [10].

As a large metropolitan area, Nelson Mandela Bay (NMB) has a variety of environmental health hazards. Agunbiade, Adeniji, Okoh, and Okoh (2023) [11] highlighted that a number of studies have indicated that in NMB, community practices and industry are the origin of pollution in certain water sources, such as the Chatty River, Markman, and Motherwell Canal, as well as in the Swartkops estuary. Another study highlighted that traditional healers who used the river for traditional ceremonies indicated that they had to bring water from home, as the river water was too dangerous to drink [12]. Another study in NMB found that children from low-income areas showed symptoms of asthma, rhino conjunctivitis, and eczema and these were largely related to them living in homes that consisted of poor hygienic and socio-economic conditions where dampness is prevalent [13]. Such housing conditions also support the growth and reproduction of mites and mold, which increases susceptibility to allergens.

The South African National Department of Health (NDoH) [14] states that to create awareness and prevent and reduce health risks associated with environmental hazards, there is a need for inter-sectoral collaboration and community participation. Such collaboration involves the primary health care (PHC) system together with ward-based PHC outreach teams (WBPHCOTs), which include six to ten community health workers (CHWs), one data-capturer, and one (nurse) outreach team leader (OTL). The CHWs work in the district health system where they seek to promote health; prevent disease; provide therapeutic, rehabilitative, and palliative functions; and are supported by health care providers, as well as environmental health practitioners (EHP) [14]. The WBPHCOTs include generalist CHWs, who are led and supported by nurses, as well as EHPs and health promoters [15].

CHWs play an important role as a link between the formal clinic system and the people in the community. Hence, they are expected to take responsibility for a designated number of households and to form close ties with the local clinics [15]. The role of the CHW is primarily that of health promotion, prevention, and screening [16]. Traditionally, they focused on HIV and TB, but their scope now includes maternal and child health and chronic non-communicable disease care, with an emphasis on prevention, thus complementing the existing system of care and support [17]. CHWs also serve as health educators and reinforce basic health education on disease prevention and the management of chronic disease [18].

In a personal communication with the training facilitator of CHWs in the NMB, it was found that environmental health was not included in the training modules for the CHWs. There were indirect references to environmental health in the modules, but there was no specific environmental health module. The lack of knowledge of environmental health hazards and poor practices among CHWs was highlighted by Hangulu and Akintola (2017) [19] in relation to health care waste management, leading them to suggest that CHWs needed additional training in this regard. In addition, Masilela and Olvitt (2017) [20] stated that CHWs could not differentiate between healthcare risk waste from the other waste, and their knowledge concerning its risks was limited and they did not know how to manage such waste. There is literature regarding CHWs’ attitudes towards certain aspects of their scope of practice [21], but little has been discussed regarding their attitudes and practices towards environmental health hazards. The purpose of this study was to determine the knowledge, attitudes, and practices of CHWs in relation to environmental health hazards in NMB.

## 2. Materials and Methods

### 2.1. Study Design

This study had a quantitative, cross-sectional design. It took place in NMB (Figure 1), which is a metropolitan city in the eastern half of the Eastern Cape Province with an estimated population of 1.3 million people [22]. The study site was chosen owing to the high number of CHWs working in the NMB. Data collection took place from the 6th of June to the 19th of July 2023. The NMB is divided into: sub-district A, sub-district B, and sub-district C [23], with 51 clinics and community health centers [24].

The study population consisted of all CHWs working in the NMB. The target population consisted of CHWs in the NMB that provided home-based care services to their clients in the NMB and those affiliated with PHC clinics in NMB. The study only included those who were currently active (had been working in the last three months at the time of data collection). It also included all CHWs no matter their age, gender, culture, level of education, or disability. The study excluded CHWs (including lay counsellors) who did not do home visits (as they would not engage with the environment where their clients live). It also excluded all CHWs under the age of 18, as well as those who were not affiliated with DoH clinics, as they are not under the guidance of a clinic nurse. The overall sample size was calculated using Epi Info’s StatCalc calculator (version 7.0). The sample was drawn from a population of approximately 364 CHWs [24] and was calculated to be 189 respondents (63 respondents per sub-district) based on a confidence level of 95%. Stratified random sampling was used to ensure all the sub-districts in the health district were fairly represented in the sample. Time and financial reasons prevented the researchers from including all the CHWs in the study. Not all the clinics had the same number of CHWs per clinic, so in the end, there were seven clinics from sub-district A, eight clinics from sub-district B, and seven clinics from sub-district C, amounting to 22 clinics in total. Random sampling was used to select the clinics from each sub-district using a digital name generator. The selection of participants was in the form of a census, as all CHWs at each clinic were invited to take part and sufficient questionnaires were distributed to ensure all the current CHWs had a chance to participate. Altogether, 179 questionnaires were distributed and the response rate of completed questionnaires was 61% or 110 out of 179. According to Fekete, Segerer, Gemperli, Brinkhof, and SwiSCI Study Group (2015) [25], 60% has largely been accepted as a sufficient response rate for a survey.

### 2.2. Measurements, Quality Control, and Pre-Test Study

The questionnaire was based on the literature [26,27], consisting of 36 items in four sections. Section A consisted of demographic data concerning the CHWs, namely, age; gender; education level; courses undertaken; years of experience as a CHW; and the sub-district where they were based. Section B, C and D consisted of Likert scales and are aligned to the purpose of this study, which addresses the CHWs’ knowledge, attitudes, and practices towards environmental health hazards. Section B consisted of knowledge questions regarding environmental health hazards with the option of answering: “True”, “False”, and “Unsure”. The questionnaire included ten knowledge questions about environmental health hazards and the maximum score attainable was 10 out of 10. Respondents who obtained a knowledge score of 9 or 10 out of 10 were classified “Excellent”. Those who got 7 or 8 were classified “Good”. Those who obtained 5 or 6 were classified “Moderate”. Those who achieved 3 or 4 were classified “Poor”, while those who obtained 1 or 2 were classified “Very poor”. 

Section C addressed the attitudes of CHWs regarding environmental health hazards and consisted of “Strongly Agree”, “Agree”, “Unsure”, “Disagree”, and “Strongly Disagree”. The questionnaire included ten attitude questions about environmental health hazards. Respondents who obtained an attitude score of 9 or 10 out of 10 were classified as having a “Very Positive Attitude”. Those who got 7 or 8 were classified as having a “Positive Attitude”. Those who obtained 5 or 6 were classified as having a “Moderately Positive Attitude”. Those who had 3 or 4 were classified as having a “Negative Attitude”, while those who obtained 1 or 2 were classified as having a “Very Negative Attitude”. 

Section D ascertained the practices of CHWs regarding environmental health hazards and consisted of “Always”, “Sometimes”, and “Never”. The questionnaire collected data on CHW practices in relation to environmental health hazards. Ten statements were presented regarding CHW practices, requesting the respondents to indicate if they always did the particular practice, whether they sometimes did it, or whether they never did it.

A pre-test on eight CHWs was conducted prior to conducting the main study at a clinic that was not part of the main study. Together with the pre-test, the review process of the research and ethics committees helped enhance the face and content validity of the questionnaire. No changes were made to the questionnaire. However, it was decided to include an isiXhosa translation to accompany the English translation to assist the CHWs with completing the questionnaire to assist the respondents to provide more accurate answers.

Regarding reliability, after the completion of the study, the Cronbach alpha for the questionnaire was assessed. Regarding the internal consistency of the questionnaire, the Cronbach alpha for attitudes was 0.9 and for practices, it was 0.83. As the knowledge questions were true and false questions requiring a right or wrong answer, a Cronbach alpha was not applicable.

During recruitment, the respondents received information letters and consent forms from the fieldworkers who explained the information letter to the CHWs. The fieldworker (or gatekeeper) explained to the CHWs who chose to take part how to complete the questionnaire. Questionnaires were all anonymous with no names attached to them. They were inserted into an envelope marked “Questionnaires” and the fieldworker collected the questionnaires a few days after distribution.

### 2.3. Data Analysis

The analysis was mainly descriptive statistics, such as frequencies and summary statistics. Group comparisons with independent sample *t*-tests were also conducted. Three variables were assessed: knowledge, attitudes, and practices.

### 2.4. Ethical Considerations

This study received ethical clearance from the University of Johannesburg Faculty of Health Sciences Research Ethics Committee (clearance number: 1636–2022) and Eastern Cape Provincial Department of Health Research Ethics Committee (reference number: EC_202208_003). All persons who participated in this study provided written consent prior to taking part and participation was voluntary. Privacy and confidentiality were maintained throughout the interviews and questionnaires were filled in anonymously.

## 3. Results

### 3.1. Demographic Characteristics of Study Participants

Most of the respondents, 94.5% (*n* = 104), were female, with the rest, 3.6% (*n* = 4), being male. The mean age of the respondents was 44 years. Most of the respondents, 51.8% (*n* = 57), indicated Grade 12 to be their highest qualification, followed by 32.7% (*n* = 36) in the Grade 10–11 category. Hence, 84.5% (*n* = 93) respondents had an education level between Grade 10 and 12. Regarding training in environmental health, 62.7% (*n* = 69) of the respondents said “yes” they had received some education in environmental health. Conversely, over a third of the respondents, 34.5% (*n* = 38), had not received any training in environmental health. Regarding the length of time working as a CHW, the mean score for the sample was approximately six years per CHW.

### 3.2. Knowledge of Environmental Health Hazards

A small percentage of respondents had “Excellent” knowledge scores, 4.55% (*n* = 5), and a slightly larger percentage had “Good” scores, 17.27% (*n* = 19) (Figure 1). Most of the respondents had a “Moderate” score, 57.27% (*n* = 63). Altogether, 16.36% (*n* = 18) had a “Poor” score and 4.55% (*n* = 5) had a “Very Poor” score. Hence, on the extremes, 21.82% had “Excellent” or “Good” scores, while 20.91% had “Poor” or “Very Poor” scores. Therefore, 79.09% (*n* = 87) of the respondents had moderate to excellent knowledge of environmental health hazards based on this questionnaire (Figure 2).

Most of respondents indicated that environmental health hazards are usually easy to spot. However, this answer was false and only 14.7% (*n* = 16) of the respondents answered correctly. Most of the respondents felt that “A person’s environment is less important than his or her genetics when it comes to determining risk for disease”. However, this answer was false and only 24.8% (*n* = 27), or a quarter, of the respondents answered correctly. Half of the respondents indicated that “Air pollution is found only outdoors”. The answer here was false and half the respondents, 50% (*n* = 55), gave the incorrect answer to this question. The respondents were asked if an individual’s work environment could increase their risk for cancer, and 50.5% (*n* = 55) said “no”, which was the incorrect answer.

### 3.3. Attitudes towards Environmental Health Hazards

A quarter of the respondents had a “Very Positive Attitude”, 25.45% (*n* = 28). Over a third of the respondents, 34.55% (*n* = 38), had a “Positive Attitude” (Figure 2). A small percentage, 13.64% (*n* = 15), had a “Moderately Positive Attitude”. Less than a fifth had a “Negative Attitude”, 18.18% (*n* = 20), while 8.18% (*n* = 9) had a “Very Negative Attitude”. As such, 73.64% (*n* = 81) of all the respondents had a “Moderately Positive” to “Very Positive Attitude” towards addressing aspects of environmental health hazards (Figure 3).

Over a third of the respondents, 37.38% (*n* = 40), agreed that CHWs were not responsible for environmental health hazards. Altogether, 37% (*n* = 40) of the respondents agreed that environmental health is not a part of the practice of health promotion. Hence, over a third of the CHWs in this study believed there is no relationship between environmental health and health promotion. Most of the respondents, 62% (*n* = 67), disagreed that they did not need additional training in environmental health. Most of the respondents, 61.5% (*n* = 67), disagreed that environmental health is the responsibility of the environmental health practitioner only. However, over a quarter of the respondents, 28.4% (*n* = 33), agreed with the statement. Altogether, 28.7% (*n* = 31) of the respondents agreed that they cannot make a difference to the health of the home environment of their clients. However, 72.2% (*n* = 78) of the CHWs felt that they were able to make a difference to the health of the home environment of their clients. Over a third of the respondents, 39.4% (*n* = 43) agreed that their fellow CHWs know very little about environmental health hazards.

### 3.4. Practices in Relation to Environmental Health Hazards

Nearly a quarter of the respondents, 24.1% (*n* = 26), stated that they only sometimes check each household for environmental hazards. Over a quarter of respondents, 28% (*n* = 31), said they only sometimes warned households of the dangers of mold. Most of the respondents, 64.2% (*n* = 70), indicated that they always warn households about rubbish and rats. It is positive that the majority indicated that they always warned households of this hazard. However, 28.4% (*n* = 31), or just over a quarter, of respondents stated that they only sometimes warn households about rubbish and rats, and this suggests that perhaps they do not fully understand the dangers of vermin and their ability to spread disease. The majority of respondents, 68.2% (*n* = 75), indicated that they always tell households about the dangers of damp. However, a quarter, 24.5% (*n* = 27), of the respondents stated that they only sometimes tell households about the dangers of damp. Most of the respondents, 71.6% (*n* = 78), said that they always warn households about the dangers of indoor pollution. However, 23.9% (*n* = 26) of respondents said they only sometimes did this. Furthermore, the response to this item does appear to contradict the knowledge question, where half the respondents did not appear to know the dangers of indoor air pollution. The majority of respondents, 62.4% (*n* = 68), indicated that they always reported environmental hazards to the clinic. However, over a quarter, 28.4% (*n* = 31), said they sometimes did this, while 9.2% (*n* = 10) said they never did this.

### 3.5. Group Comparisons

Independent sample *t*-tests were performed to determine whether there was a significant difference in the mean knowledge, attitudes, and practices dependent on whether the respondents were trained in environmental health. It was found that there were significant differences in attitudes (t = −2.308, df = 91.107, *p* = 0.023) and practices (t = −2.936, df = 62.491, *p* = 0.005). The results show that those who had been trained in environmental health had a significantly lower mean attitudes score (higher level of agreement) (m = 3.2365, sd = 1.113) compared to those who did not have training in environmental health (m = 3.694, sd = 0.894). In addition, it was also found that those who had been trained in environmental health had a significantly lower mean practice score (practiced more frequently) (m = 1.231, sd = 0.327) compared to those who did not have training in environmental health (m = 1.4605, sd = 0.4162).

## 4. Discussion

Almost all the CHWs were female, with an average age of 44. This appears to be in alignment with another study [28], where 95% of the CHWs were female and the median age was 43. Over a third of the respondents had not received training in environmental health. A United States study [29] found that training in environmental health was not common among the CHWs, with over half not having had training in environmental health. A 2018 review [17] highlighted that CHW training and the scope of work of CHWs had not given sufficient attention to the social determinants of health and there has been inadequate support from EHPs. This suggests that the environmental health training provided to CHWs has been relatively limited in recent times. The mean score for years of experience among the CHWs was six years. In a KwaZulu-Natal study [30], it was found that out of the 53 participants, the level of experience as a CHW was 7.7 years, while 82% (*n* = 91) of the CHWs in another study [31] had worked as CHWs for more than five years. It appears that CHWs tend to remain in the position as CHW for an extended period of a time and, therefore, investing in their education, particularly regarding environmental health, will benefit the PHC system.

### 4.1. Knowledge

There was a general misconception among the CHWs that environmental hazards are easy to identify. Many hazards are not easy to identify and there are many challenges and difficulties with hazard identification [32]. A hazard such as contaminated food is not easy to identify because the bacteria in food are not visible to the naked eye. A study in Nelson Mandela Bay found that the chances of food contamination in informal settlements is high because access to running water is limited, homes are not adequately ventilated, and many homes still rely on a bucket system [33].

Most of the CHWs indicated that the environment was less important when determining disease risk. However, the environment in which people live plays a major role in determining whether an individual is at risk of disease. Many cancers are caused by environmental toxicants or carcinogens [34]. Indeed, gene–environment interaction shows that genes, as well as exposure to physical, chemical, biological, and psychosocial factors, are the cause of many cancers [35]. Hence, there appears to be a knowledge gap among the CHWs regarding the relationship of the environment to disease.

Half of the CHWs did not appear to be aware of indoor air pollution. Indoor air pollution is a major a cause of respiratory conditions, particularly among infants, and almost doubles the risk for childhood lower respiratory conditions and causes 44% of all pneumonia deaths in children under five in low- and middle-income countries [36,37]. Many South African homes continue to utilize wood fuel, gas, coal, and paraffin for household purposes, and many others with access to electricity use wood and coal for cooking and heating because of the high costs of electricity [38]. Other conditions, such as asthma, are also exacerbated by indoor air pollution.

### 4.2. Attitudes

More than a third of the CHWs felt that CHWs were not responsible for environmental health hazards. The scope of work of CHWs requires them to “conduct community, household and individual-level health assessments” [17]. Such assessments require them to be equipped to monitor and report environmental health hazards to the relevant stakeholders [39]. In addition, based on the above response, there does appear to be a need to find ways to develop relationships between CHWs and EHPs in “undertaking/catalyzing local environmental activities” [17].

A third of the CHWs felt that environmental health is not a part of the practice of health promotion. Hence, some CHWs do not appear to have a clear understanding of the link between environmental health and health promotion. There is a strong link between health prevention/promotion and environmental systems [40] and there are links between health promotion and issues such as sanitation and hygiene, which are strongly associated with environmental health [41]. Furthermore, the National Health Promotion Policy and Strategy also highlights the importance of the relationship between health promotion and environmental health [42]. Therefore, there appears to be a training gap among CHWs regarding the relationship between environmental health and health promotion. 

Most of the CHWs indicated that they needed additional training in environmental health. A South African study [43] highlighted that there was a gap in CHW education regarding the need for regular refresher training, which links to the respondents’ desire for extra training. This is further supported by Schneider et al. (2018) [17], who stated that the training of CHWs lacked an emphasis on the social determinants of health that are an important component of environmental health training. Related to training, most of the CHWs felt that environmental health was not the sole responsibility of EHPs. In general, owing to task shifting, CHWs from many settings are not always clear about their roles and responsibilities [44]. Hence, the roles of CHWs need to be clarified and communicated to other health practitioners [45].

### 4.3. Practices

Almost a quarter of the CHWs indicated that they only sometimes check each household for environmental hazards. However, the role of CHWs includes screening households for surrounding environmental hygiene and sanitation challenges [46]. The NDoH’s [26] “Health for all” document, which is a health promotion tool for CHWs, highlights a range of environmental hazards that CHWs need to look out for on their rounds in the community.

Over a quarter of the CHWs said they only sometimes warned households of the dangers of mold. This is particularly important, considering that mold or fungi have the potential to negatively affect any person in the household with respiratory problems, especially children [47]. However, if CHWs do not regularly remind their clients of the importance of keeping mold under control, it could increase the chances of respiratory problems among children with asthma. A study of NMB [48] highlighted the problem of poor housing conditions where leaking roofs and leaking water pipes led to fungal infestations, which cause respiratory conditions.

More than a quarter of the CHWs stated that they only sometimes warn households about rubbish and rats. CHWs can play a critical role in rodent control. For instance, in a Madagascan study [49], the government utilized CHWs to circulate information regarding vermin and rodent control to the communities.

A quarter of the CHWs stated that they only sometimes tell households about the dangers of damp. A study in Nelson Mandela Bay [48] highlighted the problem of poor-quality housing in the two townships in their study with a specific emphasis on high levels of damp. The issue of dampness in homes and its association with poor health outcomes was also described in another South African study in Mpumalanga Province and North-West Province [50]. These studies emphasize the importance of CHWs being vigilant for issues of damp as an environmental hazard.

Most respondents said that they always warn households of indoor pollution. However, this appears to contradict the response to knowledge, where half the respondents did not appear to know the dangers of indoor air pollution. As mentioned, indoor air pollution harms respiratory health and may trigger allergic and irritant reactions, such as asthma [49]. The NDoH [50] states that exposure is particularly high among women and young children, who spend the most time in houses or near emission sources.

One of the limitations of the study was that there are very few studies on the topic of CHWs and environmental health. Hence, there were limited studies to help guide the development of the questionnaire. However, the results have helped to clarify how the questionnaire could be enhanced to elicit richer data. A more comprehensive set of knowledge questions would have yielded a more accurate reflection of the knowledge levels of the CHWs. The size of the pre-test sample prevented the researchers from assessing the questionnaire’s reliability prior to the main study; however, this was assessed after the completion of the study. The generalizability of the study findings was undermined by the sample size. A qualitative component could have helped unpack more about the context in which the CHWs work and how this relates to their views of environmental health hazards. A qualitative study could also unpack in greater detail the attitudes and practices of CHWs regarding environmental health hazards.

There is a need for future research that involves in-depth interviews or focus group discussions on the topic of environmental health hazards and the role of CHWs in this regard. Research involving EHPs and their views regarding the role or potential role of CHWs in assisting them with addressing environmental health challenges is also needed. Research exploring the perceptions and roles of EHPs could offer additional insights into how intersectoral collaboration could be enhanced.

## 5. Conclusions

The majority of the CHWs had a moderate knowledge score concerning environmental health hazards. Furthermore, the majority of the CHWs had a very positive or positive attitude towards environmental health hazards. Over a third of the CHWs felt they were not responsible for environmental health hazards. It appears that they did not see environmental health hazards as being a part of their scope of work. A large majority of CHWs felt that they needed additional training in environmental health. Regarding CHW practices in relation to environmental hazards, almost a quarter of the CHWs indicated that they only sometimes checked each household for environmental hazards. This response suggests that environmental health is not necessarily a priority for all CHWs. Hence, there is a need for CHWs to receive very specific training in environmental health, particularly as it relates to the kind of work that they do. There is a need to integrate environmental health training into the basic CHW curriculum and to provide ongoing training modules to improve environmental health knowledge and practices among CHWs. It is also recommended that CHWs be equipped to provide health education regarding environmental health hazards to the community members whom they serve. There also appears to be a need to find ways to develop relationships between CHWs and EHPs and to further explore the scope of work of CHWs in relation to environmental health. There is a need for greater clarity in existing CHW policies regarding the role of CHWs in relation to environmental health and, indeed, in relation to EHPs who are key role players in the WBPHCOT. Furthermore, CHW policies need to include guidelines regarding the environmental health training of CHWs.

## Figures and Tables

**Figure 1 ijerph-21-00353-f001:**
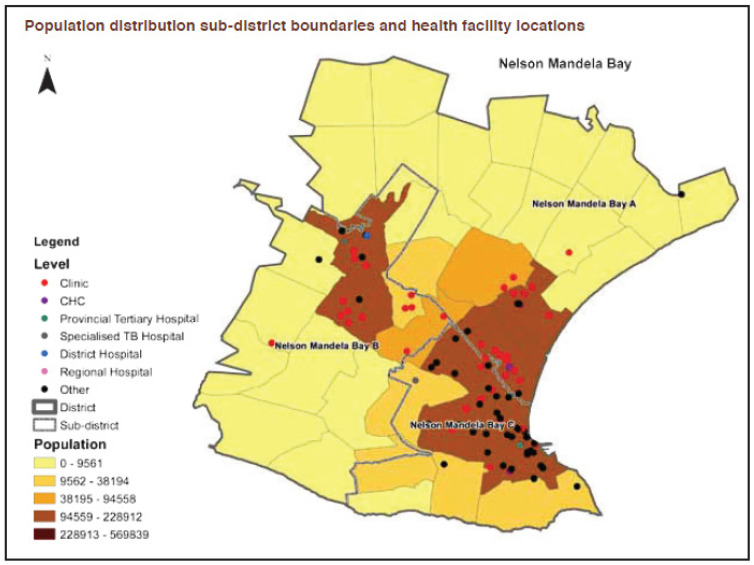
Map of NMBM with sub-districts [23].

**Figure 2 ijerph-21-00353-f002:**
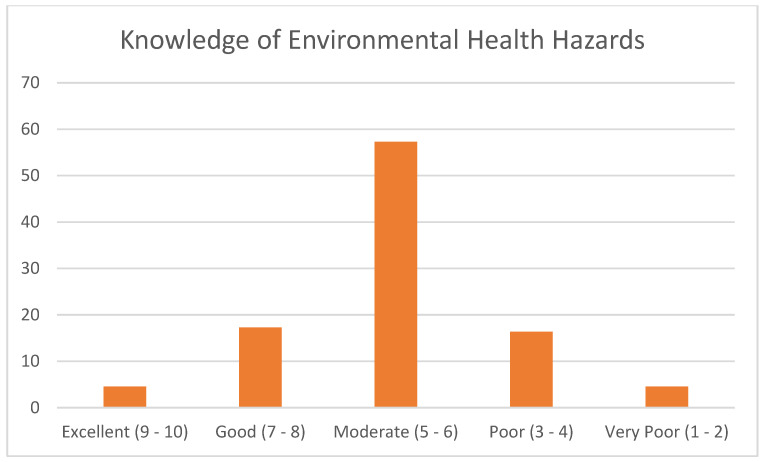
Overall knowledge scores of the CHW participants.

**Figure 3 ijerph-21-00353-f003:**
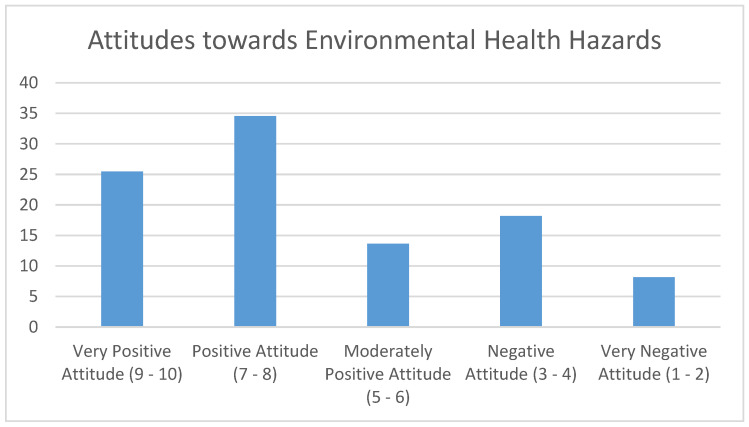
Overall attitudes scores of the CHW participants.

## Data Availability

The raw data supporting the conclusions of this article will be made available by the authors on request.

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
