# Peer review of "Knowledge, Attitudes, and Practices of Community Health Workers in Relation to Environmental Health Hazards in the Nelson Mandela Bay"

_ijerph, 2024, doi:10.3390/ijerph21030353_

Round 1

Reviewer 1 Report

Comments and Suggestions for Authors

This article uses a questionnaire survey to sample 110 community health workers, and through statistical analysis presents their views and opinions on the environmental health of Nelson Mandela Bay. The following suggestions are provided for reference by the worker group:

  First, please provide additional explanation on the reasons for choosing Nelson Mandela Bay (NMB) as the case for this study? What is special or different about it from other urban settings in South Africa?

 Second, this study already known that there are 364 CHWs in the questionnaire survey population, why not conduct a census? What are the reasons?

Third, NMB is divided into three sub-districts ( ABC), and has 51 clinics and community health centers. It is recommended that additional explanation be required regarding their distribution status, as well as the background, population, and natural environment of each district. 

Fourth, please describe the time of the questionnaire survey. 

Fifth, please provide additional explanation for the questionnaire that is not in Appendix A.

Sixth, what are the results of the pilot study? Recommended statement, including reliability and validity of the questionnaire. 

Seventh, this study also conducted factor analysis, including t-test, correlation and group comparison. Please explain which items these statistical analysis methods were used on, because many of the results presented in the text are only presented in the form of descriptive statistics. , such as sample numbers and percentages.

Author Response

Responses are in blue color.

Reviewer 2 Report

Comments and Suggestions for Authors

Abstract

·       Pls mention the standard questionnaire or instruments (if any) that were used in this study in method section.

·       Present specific values for significant results instead of just percentages in the results.

·       The conclusion is too general. “It appears that environmental health is not a priority for all CHWs or rather that their focus might be elsewhere”. Pls be more specific and based on the results.

Introduction

·       Acknowledge the well-explained introduction supported by citations.

Method

Study design

·       From this statement, “The target population consisted of CHWs in the NMB that provided home-based care services to their clients in the NMB and affiliated to PHC clinics in NMB, but not those affiliated to NPOs”, Pls justify why this population were selected? Justify the selection criteria of CHWs in the NMB and provide clarity on excluding those affiliated with NPOs.

·       How about the exclusion and inclusion criteria of this study? Pls list out and justify.

·       What does the author mean by draft questionnaire? Is that a self-administered questionnaire?

Measurements, Quality Control, and Pilot Study

  • Specify the chosen Likert scale (3-point or 4-point) for Sections B, C, and D, with justification and references. Is there any reference to support? Why different Likert point for each section.
  • Address any discrepancies regarding knowledge questions about environmental health and their absence in the pilot study.

Variables

·       The questionnaire included ten knowledge questions about environmental health 155

hazards. Respondents who obtained a knowledge score of 9 or 10 were classified “Excellent”. Those who got 7 or 8 were classified “Good”. Those who obtained 5 or 6 were classified “Moderate”. Those who achieved 3 or 4 were classified “Poor” while those who obtained 1 or 2 were classified “Very poor”. Suddenly the author mentioned about knowledge question about environmental health which was not mentioned in the pilot study section. It was very confusing to the readers. Additionally, this section seems to have separate questions and Likert scale use compared to the previous section which are so unclear and contradict each other.

Results

·       For the graph results it is suggested to put the value of the score range for each category

·       In data analysis, the author mentioned the use of Factor analysis, including t-tests, correlations, and group comparisons for the variables of Knowledge, attitudes and practices. However, seems this paper only mentioned about the percentage not more than that which I think this paper is very surface and no in-depth analysis was conducted.

·       Provide more in-depth analysis beyond percentages.

Discussion

·       The discussion is based on the results which is not details or in depth discussion and too general. Suggestion to analyse each factor since this study involve many factors to identify the contributing factors in details.

·       Move beyond a general discussion and delve into specific contributing factors based on the study's method and results.

Conclusion

  • Align the conclusion with the study's objectives, focusing specifically on environmental health hazards.
  • Summarize key findings related to environmental health without introducing new information about hazards that are difficult to recognize or classify.

Comments on the Quality of English Language

The text is clear and well-structured, making it easy to understand. Manuscript is both informative and linguistically well-executed.

Author Response

Responses are in blue color.

Reviewer 3 Report

Comments and Suggestions for Authors

 IJERPH Manuscript ID: ijerph2850204

Review and Comments on "Knowledge, Attitudes, and Practices of Community Health Workers in Relation to Environmental Health Hazards in the Nelson Mandela Bay"

After reviewing the document, here's a detailed assessment and suggestions, then questions

Title and Abstract:

-        The title reflects the study's content.

-        The abstract should include details about the study design and key findings to enhance its informativeness.

Introduction:

It adequately sets the study context but could better link CHW training in environmental health to potential community health impacts.

Methods:

-        Appropriately designed (quantitative crosssectional) for the research question.

-        The sampling and data collection descriptions are detailed, yet the rationale behind the sample size calculation needs elaboration.

-        The questionnaire design is literature-based, but a more comprehensive description of its alignment with the research objectives is recommended.

Results:

-         Clearly presented with appropriate statistics.

-         Findings show a need for additional CHW training in environmental health.

-         Suggest adding a more comprehensive analysis or discussion on the implications of these findings.

Discussion:

-         Effectively highlights key findings and implications for CHW training.

-         Could further explore the potential impact of improved training on environmental health outcomes.

-        Discuss how study limitations might affect the findings' generalizability.

Conclusion:

-         Concisely summarizes findings and their implications.

-         Suggests a need for specific training in environmental health for CHWs.

Comments and Suggestions for Authors

1.           Abstract:

-              Incorporate specific study design details in the abstract and highlight key findings to give readers a concise overview of your research.

2.           Introduction:

-              Establish a direct link between gaps in Community Health Worker (CHW) training in environmental health and their potential impacts on community health. Integrate relevant literature to enhance the background of the study.

-              Expand the introduction to provide a more comprehensive overview, setting the stage for your research.

3.           Sample Size Rationale:

-              Provide a comprehensive rationale for calculating the sample size, explaining the choice of confidence level and margin of error to ensure the reader understands the basis for your sample size determination.

4.           Questionnaire Alignment:

  Clearly articulate how the questionnaire aligns with the research objectives. Ensure the reader can see the direct connection between your research goals and the survey questions.

5.           Positive Impact of CHW Training:

-              Explore the potential positive impact of enhanced CHW training on environmental health outcomes. Elaborate on how improved training could lead to better community health outcomes.

6.           Study Limitations:

-              Address the potential influence of study limitations on the generalizability of findings. Acknowledge the limitations and their implications on the broader applicability of your research.

7.           Sentence Structure and Readability:

-              Simplify complex sentences and employ straightforward language to enhance readability throughout the paper, making it accessible to a wider audience.

8.           Results Section:

-              Present similar findings more directly in the results section for a more straightforward interpretation. Consolidate related results to improve reader-friendliness and clarity.

9.           Methodology Section:

-              Enhance the methodology section by providing more detailed information, making it useful for replication and validation of your study. Include specific steps, data collection methods, and statistical procedures.

10.        Conclusions:

-              Strengthen the conclusions by directly linking them with your findings. Discuss the broader implications of your research and its potential impact on community health. Make sure your conclusions align with the objectives and results of your study.

Questions

1.       Addressing Potential Biases in Participant Selection and Ensuring Sample Representativeness:

  How did the study mitigate potential biases in participant selection, and what measures were implemented to ensure the sample's representativeness?

2.       Assessing Validity and Reliability of Questionnaire Data:

  Was any effort undertaken to evaluate the validity and reliability of the data collected through the questionnaires?

3.       Comparing Findings with Existing Literature on CHW Training in Environmental Health:

   How do the study's findings compare with the existing body of literature regarding Community Health Worker (CHW) training in environmental health?

4.       Implications for Policy and Practice in CHW Training Programs:

  What potential implications do the study's results hold for policy development and practical implementation in Community Health Worker (CHW) training programs?

5.       Benefits of a Mixed Methods Approach, Including Qualitative Data:

  Would including qualitative data through a mixed methods approach have enhanced the study's overall quality and depth of analysis?

Overall Assessment

The manuscript addresses a crucial gap in CHW training literature. While the methodology is sound, it requires additional details on sample size justification and questionnaire alignment.

Comments on the Quality of English Language

The paper is well-written, but it could benefit from minor grammatical corrections for enhanced clarity. Simplify complex sentences and use straightforward language to improve readability throughout the paper, making it accessible to a wider audience.

Author Response

Responses are in blue color.

Round 2

Reviewer 1 Report

Comments and Suggestions for Authors

  The author group responded quickly, which is admirable. The following suggestions are provided for reference by the worker group:

First, please write down in this article the reasons why you cannot conduct a census.

Second, please provide additional explanation for the questionnaire in Appendix A.

Third, the results of the pre-test should be explained. Recommended statement, including reliability and validity .

Author Response

We have addressed all comments by reviewer 1.

Reviewer 3 Report

Comments and Suggestions for Authors

[IJERPH] Manuscript ID: ijerph-2850204

 Comments and Suggestions for the Authors

Your study makes an essential contribution to understanding the role of CHWs in environmental health. The positive attitude of CHWs towards environmental health, despite notable gaps in knowledge and practices, is particularly promising for future training initiatives. Expanding your research to include qualitative insights could provide a richer understanding of the challenges and opportunities faced by CHWs in this area. Developing targeted training modules based on your findings could significantly impact CHWs' effectiveness in addressing environmental health hazards.

Please see my comment below.

 1.     The study provides valuable insights into the knowledge, attitudes, and practices of community health workers (CHWs) regarding environmental health hazards in Nelson Mandela Bay (NMB). It highlights the need for specific training and support in this area.

2.     The findings indicating a link between training in environmental health and CHWs' attitudes and practices are significant. These insights can guide the development of future training programs.

3.     It is noteworthy that the majority of CHWs showed a positive attitude towards environmental health despite gaps in knowledge and practices. This suggests a solid foundation upon which training initiatives can be built.

Suggestions:

1.     A more detailed exploration of the reasons behind CHWs' attitudes and practices, possibly through qualitative studies, could provide a deeper understanding of the barriers and facilitators to improving environmental health-related practices.

2.     Expanding the research to include perceptions and roles of environmental health professionals could offer additional insights into how intersectoral collaboration can be enhanced.

3.     Integrating environmental health training into the basic CHW curriculum and providing ongoing training modules might be crucial for improving knowledge and practices in this critical area.

4.     The authors might consider developing specific policy recommendations or interventions based on the study's outcomes to promote environmental health among the communities served by CHWs.

Sincerely,

Reviewer

Author Response

We have addressed all comments by reviewer 3
